# Importance of consultations using mobile teams in the screening and treatment of neglected tropical skin diseases in Benin

Ronald Sètondji Gnimavo[1,2☯], Faraj Fajloun[3,4,5☯], Charbel Al-Bayssari[6], Espoir Sodjinou[1], Akimath Habib[1], Line Ganlonon[1], Eric Claco[1], Irvine Agoundoté[1], Odile Adjouavi Houngbo[1], Esaï Gimatal Anagonou[7], Chabi Alphonse Olaniran Biaou[7], Adjimon Gilbert Ayélo[7], Jean Gabin Houezo[7], Alexandra Boccarossa[3], Elie Hajj Moussa[5], Béatriz Gomez[8], Anna Gine[8], Ghislain Emmanuel Sopoh[2], Estelle Marion[3], Roch Christian Johnson[9], Marie Kempf[3,10]*

1 Centre de Dépistage et de Traitement de la Lèpre et de l´Ulcère de Buruli de Pobè, Fondation Raoul Follereau, Pobè, Bénin, 2 Institut Régional de Santé Publique- Comlan Alfred Quenum, Université d'Abomey Calavi, Ouidah, Bénin, 3 University of Angers, Nantes Université, CHU Angers, Inserm, INCIT, Angers, France, 4 Ecole Doctorale en Sciences et Technologie, Université Libanaise, Campus Rafic Hariri, Hadath, Liban, 5 Laboratoire d'Innovation Thérapeutique, Faculté de Sciences 2, Campus Pierre Gemayel, Fanar, Liban, 6 Departement of Medical Laboratory Sciences, Faculty of Health Sciences, University of Balamand, Tripoli, Lebanon, 7 Programme National de Lutte contre la Lèpre et l'Ulcère de Buruli, Ministère de la Santé, Cotonou, Bénin, 8 Fondation Anesvad, Henao, Bilbao, Spain, 9 Fondation Raoul Follereau, Health Department, Paris, France, 10 Département de Biologie des Agents Infectieux, UF de Bactériologie, Centre Hospitalier Universitaire Angers, Angers, France

☯ These authors contributed equally to this work.
* MaKempf@chu-angers.fr

**Data Availability Statement:** All epidemiological data are available in the manuscript.

## Abstract

### Context

Since 2013, the World Health Organization has recommended integrated control strategies for neglected tropical diseases (NTDs) with skin manifestations. We evaluated the implementation of an integrated approach to the early detection and rapid treatment of skin NTDs based on mobile clinics in the Ouémé and Plateau areas of Benin.

### Methods

This descriptive cross-sectional study was performed in Ouémé and Plateau in Benin from 2018 to 2020. Consultations using mobile teams were performed at various sites selected by reasoned choice based on the epidemiological data of the National Program for the Control of Leprosy and Buruli Ulcer. All individuals presenting with a dermatological lesion who voluntarily approached the multidisciplinary management team on the day of consultation were included. The information collected was kept strictly anonymous and was entered into an Excel 2013 spreadsheet and analyzed with Stata 11 software.

### Results

In total, 5,267 patients with various skin conditions consulted the medical team. The median age of these patients was 14 years (IQR: 7–34 years). We saw 646 (12.3%) patients

**Funding:** The preparatory phase and the implementation of this approach on the sites were financed by the Raoul Follereau Foundation France in collaboration with the National Program for the Fight against Leprosy and Buruli Ulcer of Benin and the by ANESVAD Foundation. RCJ received grants from Raoul Follereau Foundation France; BG received grants from ANESVAD Foundation; JGH received grants from National Program for the Fight against Leprosy and Buruli Ulcer of Benin. The funders had no role in study design, data collection and analysis, decision to publish, or preparation of the manuscript.

**Competing interests:** The authors have no competing interests to declare.

presenting NTDs with skin manifestations, principally scabies, in 88.4% (571/646), followed by 37 cases of Buruli ulcer (5.8%), 22 cases of leprosy (3.4%), 15 cases of lymphatic filariasis (2.3%) and one case of mycetoma (0.2%). We detected no cases of yaws.

## Conclusion

This sustainable approach could help to decrease the burden of skin NTDs in resource-limited countries.

## Author summary

Neglected Tropical Diseases (NTDs) with skin manifestations are a group of nine infectious diseases with skin manifestations, either as the primary symptom or as an associated clinical feature. They are often co-endemic and harbor similar clinical signs.

We evaluated the implementation of an integrated approach to the early detection and rapid treatment of skin NTDs based on consultations using mobile teams in the *départements* of Ouémé and Plateau in Benin.

In total, 5,267 patients with various skin conditions consulted. We saw 646 (12.3%) patients presenting NTDs with skin manifestations and among them, 571 (88.4%) had scabies, 37 (5.8%) had Buruli ulcer, 22 (3.4%) had leprosy, 15 (2.3%) had lymphatic filariasis and 1 (0.2%) had mycetoma.

Consultations using mobile teams constitute a very efficient strategy for the integrated screening of skin diseases and especially of NTDs with skin manifestations. This sustainable approach could help to decrease the burden of NTDs in resource-limited countries.

## I. Introduction

Neglected tropical diseases (NTDs) constitute a group of 20 prevalent infectious diseases affecting more than a billion people worldwide. They are particularly prevalent in poor countries with a tropical or subtropical climate, particularly in isolated rural areas, conflict zones and informal settlements. They are very common in areas without a safe supply of drinking water and with poor access to healthcare facilities, where people live in poor hygiene conditions, close to animals and the vectors of infectious disease [1–3]. NTDs cost billions of dollars in terms of healthcare costs and lost productivity, providing a strong economic argument for investment in their prevention, treatment and cure [4–5].

Nine of the 20 NTDs have skin manifestations, either as the primary symptom or as an associated clinical feature [6]. These diseases are Buruli ulcer, leprosy, yaws, scabies, lymphatic filariasis, mycetoma, cutaneous leishmaniasis, post-Kala azar dermal leishmaniasis and onchocerciasis [7]. Unfortunately, these conditions are often diagnosed late, in patients with visible skin deformations or physical disabilities, a poor functional prognosis and alterations to quality of life. The victims of these skin NTDs are subjected to societal stigmatization throughout their lives, potentially resulting in total destitution, requiring them to beg, and significant psychological damage [8,9].

Skin NTDs have long been endemic in West African countries, including Nigeria, Benin, Ghana, Côte d'Ivoire, Burkina Faso and Togo in particular, and encouraging progress has been made towards combating them [10–12]. The World Health Assembly, held in May 2013, and the World Health Organization (WHO) Regional Committee for Africa, held in

September of the same year, both advocated the creation of NTD management programs [13,14]. The following year, the WHO African Region published its regional strategy for NTDs, in which one of the targets identified was the elimination of leprosy, control of Buruli ulcer and eradication of yaws. As a means of attaining this objective, WHO African Region requested the implementation of national NTD plans and integrated NTD programs at national level in all countries in which these diseases were endemic [10, 11].

Skin NTDs are common in Benin, a country in which both Buruli ulcer and leprosy are endemic [4]. Most of the other skin NTDs are also frequently detected in Benin [15,16]. Coordination to combat these conditions was established in 2017 in Benin. Efforts were coordinated by the National Program for the Control of Leprosy and Buruli Ulcer (PNLLUB), composed of Buruli ulcer treatment Centers (CDTLUB–*Centre de Dépistage et de Traitement de la Lèpre et de l'Ulcère de Buruli*) and Leprosy Treatment Centers (CTAL—*Centre de Traitement Anti-Lèpre*). This system relies on nursing supervisors and community relays working in collaboration with the CDTLUBs and CTALs.

This national health program has established a surveillance system for the detection, notification and management of cases by several diagnosis and treatment centers in each *département* (an administrative area equivalent to a county) of the country. The epidemiological data from this surveillance system revealed high detection rates of Buruli ulcer and leprosy in the *départements* of Ouémé and Plateau [17,18]. Based on these observations, consultations using mobile teams (mobile consultations) for the integrated screening of skin diseases, including skin NTDs in particular, were performed in these areas to decrease the burden of morbidity. These consultations made it possible to screen people and to detect skin diseases, including NTDs, earlier, and to provide treatment to prevent progression and deleterious complications.

The goal of this study is to share the lessons learned during this experience of integrated management of skin NTDs in rural districts of Benin.We report here the results of this integrated approach involving mobile consultations in Ouémé and Plateau in Benin, in 2018, 2019 and 2020.

## II. Framework and study methods

### 2.1. Implementation framework

We used an approach based on free mobile clinics in the *départements* of Ouémé and Plateau in Benin. This approach was initiated jointly by the National Leprosy Control Program and the Raoul Follereau Foundation of France. A multidisciplinary medical team (nurses, dermatology/leprosy nurses, a biomedical engineer, general practitioner, dermatologist, and public health specialist) from the CDTLUB in Pobè was responsible for performing these consultations under the supervision of PNLLUB managers.

### 2.2. Study Methods

**Ethics approval and consent for participation.**   Authorization for data collection for this study was obtained from the National Health Research Ethics Committee (Reference no. 21/MS/DC/SGM/DRFMT/CNERS/SA). These mobile consultations were part of a broader study of the factors contributing to the performance of primary and secondary prevention activities for skin NTDs in Benin covered by ethics committee authorization no. 0478/CLERB/UP/P/SP/R/SA.

On site, the objectives of the survey were clearly explained to the participants and the questionnaires were administered only free and informed verbal consent had been obtained. Confidentiality was guaranteed for the information provided, which was collected solely for the purposes of the study and will remain anonymous. We confirm that all the methods used in

this observational study complied with all research guidelines and regulations in force. All the experimental protocols involving humans described here were performed in accordance with national/international/institutional guidelines and the Declaration of Helsinki. For the children enrolled in this study, free verbal consent was obtained from their parents or guardians.

**2.2.1. Preparatory phase.** The preparatory phase involved writing letters to political and administrative authorities to explain the context, objectives and the expected results for this activity. We then trained health workers (head nurses from peripheral healthcare facilities, laboratory technicians), community relays, village volunteers and teachers. This integrated training focused on basic epidemiology, elements of positive and differential diagnosis, complications, principles of care and strategies for fighting the principal neglected tropical diseases with skin manifestations found in these two areas (Buruli ulcer, leprosy, yaws) according to data from the National Program for the Fight against Leprosy and Buruli Ulcer of Benin.

Religious leaders, local elected officials, rulers, town criers, megaphonists and village chiefs from the various sites were also briefed about these diseases to raise their awareness, in discussion sessions focusing on the epidemiology of the disease, the signs and symptoms useful for early diagnosis and, in particular, the availability of free treatment for patients suffering from these diseases attending the mobile clinics.

Social mobilization was ensured by town criers and megaphonists charged with the mission of informing the community before the arrival of the medical team and on the day of the activity.

Radio broadcasts also went out in local languages on community radio stations (*Le Roi de la Vallée* in Ouémé, and *Olokiki* and Alakétou in Plateau) to raise listeners' awareness about the existence of these diseases, their signs and early symptoms, the availability of free treatment and the timetable for the series of mobile consultations scheduled for the population, whose well-being was the priority at the heart of the initiative. This primordial step made it possible to inform the community, to prepare the community for this initiative so as to obtain its agreement, commitment, adherence, involvement and active participation, to ensure the success of this intervention.

**2.2.2. Implementation of the phase of activity.** This cross-sectional, descriptive study that took place in the *départements* of Ouémé and Plateau in Benin in 2018 (November 6 to December 20), 2019 (May 2 to August 17) and 2020 (December 1 to 22). The primary target population consisted of all individuals with dermatological lesions residing in Ouémé and Plateau during the period of in which this intervention took place.

All individuals with a dermatological lesion voluntarily presenting for consultation and examination by the multidisciplinary medical management team on the day of the consultation from whom free verbal consent for treatment was obtained were included. Those who refused to be seen were not included.

**2.2.3. Sampling methods and techniques.** Given the limited resources allocated to the fight against skin NTDs and based on the epidemiological data available from the National Leprosy and Buruli Ulcer Control Program of Benin [19], we used a non-probabilistic reasoned-choice method to select the sites for the implementation of these consultations in Ouémé and Plateau.

Finally, we used a technique of choice by convenience to select all individuals presenting dermatological lesions presenting themselves voluntarily to the multidisciplinary management team on the day of the consultation. All these people were carefully examined in a room lit by daylight, in strict respect of human dignity and in accordance with the requirements of the medical ethics code in force in the Benin [20]. Samples were collected from all suspected cases of Buruli ulcer and yaws, after verbal consent for biological confirmation had been obtained from the patient. In cases of suspected BU, swabs, taken from the undermined edges of the

ulcer, or fine-needle aspiration samples were collected and sent to the CDTLUB laboratory in Pobè for biological confirmation by polymerase chain reaction (PCR) for IS2404 [21,22]. For the confirmation of yaws, samples from suspected cases were subjected to a non-treponemal and treponemal rapid diagnostic test (DPP Syphilis Screen and Confirm Assay, Chembio Diagnostic Systems, Medford, NY, USA). Leprosy was diagnosed essentially on clinical grounds, in accordance with the recommendations of the WHO Expert Committees on Leprosy [23].

In the absence of video dermatoscopy, a diagnosis of scabies was retained on the basis of the alternative level B consensus criteria for the clinical diagnosis of scabies diagnosis of the International Alliance for the Control of Scabies [24].

**2.2.4. Study tools and variables.** An integrated data form was used to collect information on skin NTDs. On this form, we recorded sociodemographic data (age, sex, main occupation or profession of the respondent, level of education, ethnicity, religion, name of the village, district and community of origin of the patient) together with geographic coordinates and information about the socioeconomic status of the respondents. This last parameter was according to the principal component analysis (PCA) method described by Filmer et al [25].

The clinical characteristics of the dermatological lesions, the nature of the tests performed and of the samples taken, the results of the tests, the diagnosis attributed and the course of action of the management team were also recorded.

**2.2.5. Data processing and analysis.** After data collection, the forms were manually processed to check the completeness and consistency of the data collected. A careful review and sorting of the data was also done to look for duplicates. All of these steps allowed us to reassure of the quality of the collected data. In the absence of any inconsistencies, the data were entered into an Excel 2013 spreadsheet and analyzed with Stata 11 software.

The variables obtained included descriptions of the epidemiological, clinical and biological characteristics of cases. Proportions were calculated for the qualitative variables. Quantitative variables are expressed as means, with the standard deviation also indicated for those following a normal distribution. The median and range were determined for quantitative variables with a skewed distribution.

**2.2.6. Mapping.** Maps were generated with ArcView GIS 3.2 software. For each confirmed case of skin NTDs, the geographic coordinates were noted and projected onto the background map downloaded free of charge from the DIVA GIS website (https://www.diva-gis.org/). Datas for the administrative boundary were provided by Ministry of Territory Planning of Benin (https://gadm.org/download_country_v3.html).

# III. Results

## 3.1. General characteristics of the individuals with skin diseases

In total, 5,267 people consulted for skin diseases at the mobile clinics during the three study periods (**Fig 1**). During the first period (November 6 to December 20, 2018), 607 people were seen in Ouémé and Plateau. During the second period (May 2 to Aug 17, 2019), we saw 2,444 people in the two *départements*. During the third period (Dec 1 to Dec 21, 2020) 2,216 people were seen, exclusively in Plateau. The various administrative areas and districts visited by the mobile consultation team during the three periods are presented in **Table 1 and Fig 2**. The main characteristics of the patients are summarized in **Table 1**.

## 3.2. Diagnosis of neglected tropical skin diseases

In total, 648 (12.3%) patients seen at mobile consultations were diagnosed with a skin NTD. Skin NTDs accounted for 6.6% (40/607) of all the skin diseases diagnosed in 2018, 6.1% (138/

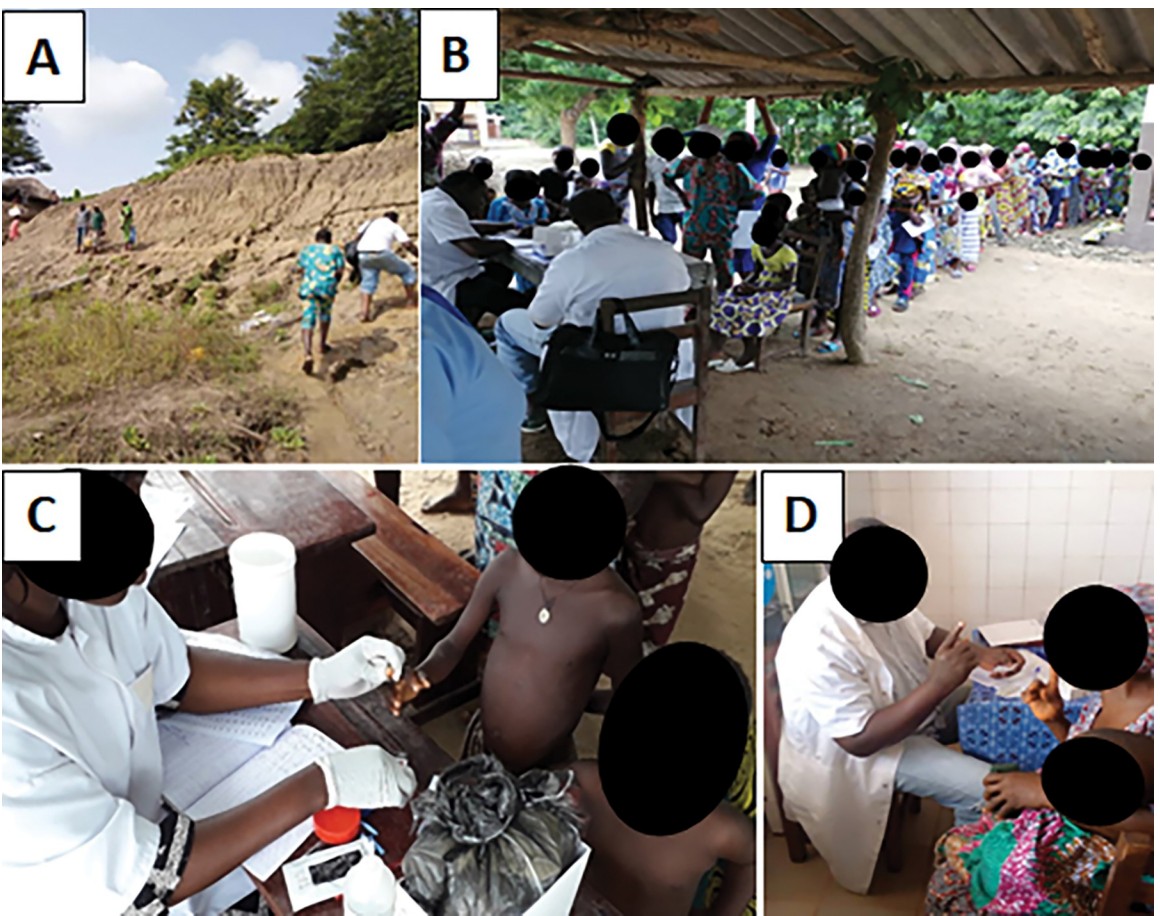

**Fig 1. Organization of mobile consultations (preparatory and implementation phase) in Ouémé and Plateau in Benin, from 2018 to 2020. A:** Multidisciplinary team from the CDTLUB of Pobè going on foot (with equipment and consumables carried in the hand and on the head in a marshy and flood-prone region, whose soil and dwellings are located at altitude) to Gnanwizoumin, a village in the Department of Ouémé, Municipality of Bonou, Arrondissement of Daméwogon, which benefited from this activity organized by the multidisciplinary team of the CDTLUB of Pobè, on 06/06/19. **B:** Reception, registration of patients (collection of socio-demographic information), taking of constants by the healthcare teams of the CDTLUB de Pobè in Attanchoukpa, a village in the Plateau department, selected to host the implementation of mobile consultations in the Commune of Kétou, District of Odométa on 08/02/20. **C:** realization by a health worker from the CDTLUB (Biomedical Engineer) of Pobè of rapid antigenic diagnostic tests for screening of yaws in children under 15 years old living in Towé, a locality in the Plateau department located in the commune of Pobè and selected for the organization of fairground consultations on 12/12/2020. **D:** Medical prescription followed by an explanatory session of the skin dermatosis and the medical prescription to a mother whose child has been received and examined by a health worker from the multidisciplinary care team of the CDTLUB of Pobè as part of the fairground consultation organized on 11/12/20 in Houédamé, a village in the Plateau department, located in the commune of Adja-Ouèrè, district of Kpoulou.

2,244) of those diagnosed in 2019, and 21.2% (470/2,216) of those diagnosed in 2020 (**Table 2**). The sites of the mobile consultations are presented in **Fig 3**. Scabies was, by far, the most frequent skin NTD, accounting for 72.5% (29/40), 66.7% (92/138) and 95.7% (450/470) of all cases of skin NTDs in 2018, 2019 and 2020. The other skin NTDs diagnosed were Buruli ulcer, leprosy, lymphatic filariasis and mycetoma.

In total, 37 cases of Buruli ulcer were diagnosed during the three periods of mobile consultations, accounting for 5.7% of the skin NTDs and 0.7% of all skin diseases diagnosed (**Fig 4A**). All these cases were confirmed by PCR. Characteristics of patients presenting Buruli ulcer are presented in **Table 3**.

During the first period of mobile consultations, three patients (7.5% of the skin NTD cases) tested positive for Buruli ulcer: two male and one female. All cases were systematically treated

**Table 1. Characteristics of the individuals with skin diseases seen in Plateau and Ouémé by the mobile consultation team during the three periods of consultations.**

| | Period 1<br>(Nov 6 to Dec 20, 2018) | Period 2<br>(May 2 to Aug 17, 2019) | Period 3<br>(Dec 1 to 21, 2020) |
|---|---|---|---|
| **Number of cases of skin diseases identified** | 607 | 2,444 | 2,216 |
| **Median age (range)** | 15 y (1–82) | 16 y (1–95) | 12 y (1–75) |
| **Sex ratio (M/F)** | 1.003 (304/303) | 0.88 (1,147/1,297) | 0.93 (1,067/1,149) |
| **Profession, no. (%)** | | | |
| Student/pupil | 261 (43.0) | 862 (35.3) | 1,053 (47.5) |
| Tradesperson | 82 (13.5) | 432 (17.8) | 181 (8.2) |
| Farmer | 79 (13.0) | 384 (15.7) | 271 (12.2) |
| No profession | 74 (12.2) | 354 (14.5) | 274 (12.4) |
| Artisan | 66 (10.9) | 238 (9.7) | 205 (9.3) |
| Household | 35 (5.8) | 139 (5.7) | 197 (8.9) |
| Civil servant | 10 (1.7) | 35 (1.4) | 35 (1.6) |
| **Municipality, no. (%)** | Pobè 230 (37.9)<br>Kétou 138 (22.7)<br>Adja-Ouèrè 82 (13.5)<br>Ifangni 82 (13.5)<br>Dangbo 73 (12.0)<br>Other 2 (0.3) | Dangbo 1,215 (49.7)<br>Adjohoun 716 (29.3)<br>Bonou 420 (17.2)<br>Adja-Ouèrè 57 (2.3)<br>Other 36 (1.5) | Pobè 1,039 (46.9)<br>Adja-Ouèrè 610 (27.5)<br>Kétou 562 (25.4)<br>Other 5 (0.2) |
| **Ethnic group, no. (%)** | | | |
| Wémé and related | 37 (6.1) | 2,251 (92.1) | 349 (15.8) |
| Holly | 106 (17.5) | 20 (0.8) | 1,081 (48.8) |
| Nagot and related | 209 (34.4) | 19 (0.8) | 705 (31.8) |
| Fon and related | 255 (42.0) | 127 (5.2) | 80 (3.6) |
| Torinu and related | - | 27 (1.1) | 1 (0) |
| **Educational level, no. (%)** | | | |
| Educated | 271 (44.7) | 897 (36.7) | 1,088 (49.1) |
| Not educated | 336 (55.4) | 1,547 (63.3) | 1,128 (50.9)) |
| **Socioeconomic level, no. (%)** | | | |
| High | - | 5 (0.2) | 22 (1.0) |
| Medium | 79 (13.0) | 160 (6.6) | 177 (8.0) |
| Low | 528 (87.0) | 2,279 (93.3) | 2,017 (91.0) |

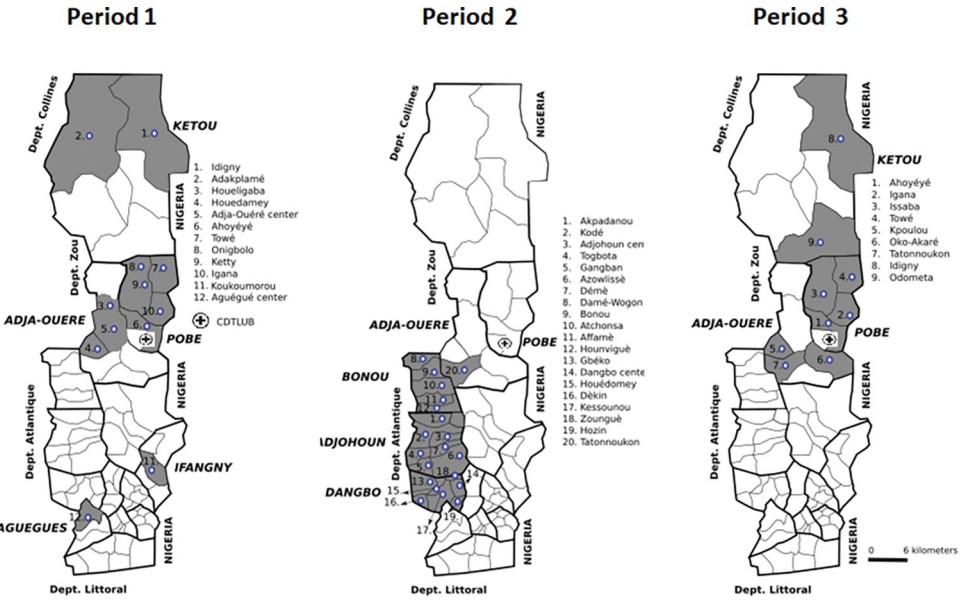

**Fig 2. Districts visited during the three periods of consultations.** Datas for the administrative boundary were provided by Ministry of Territory Planning of Benin (https://gadm.org/download_country_v3.html).

**Table 2. Results of mobile consultations for skin NTDs in Plateau and Ouémé in Benin.**

| Consultation period | Period 1 (Nov 6 to Dec 20, 2018) No. of cases—(%)–detection rate for 10,000 inhabitants | Period 2 (May 2 to Aug 17, 2019) No. of cases—(%)–detection rate for 10,000 inhabitants | Period 3 (Dec 1 to 21, 2020) No. of cases—(%)–detection rate for 10,000 inhabitants |
|---|---|---|---|
| **Total number of cases of skin diseases** | **607 (100) 29.72** | **2,244 (100) 98.58** | **2,216 (100) 121.83** |
| *Neglected tropical diseases* | *40 (6.6) 1.96* | *136 (6.1) 5.97* | *470 (21.2) 25.84* |
| Buruli ulcer | 3 (7.5) 0.14 | 28 (20.6) 1.23 | 6 (1.3) 0.33 |
| Leprosy | 6 (15.0) 0.29 | 5 (3.7) 0.22 | 11 (2.3) 0.6 |
| Scabies | 29 (72.5) 1.42 | 92 (67.6) 4.08 | 450 (95.7) 24.74 |
| Lymphatic filariasis | 2 (5.0) 0.098 | 10 (7.4) 0.44 | 3 (0.6) 0.16 |
| Mycetoma | 0 | 1 (0.7) 0.044 | 0 |
| Yaws | 0 | 0 | 0 |
| Cutaneous leishmaniasis | 0 | 0 | 0 |
| Post-Kala azar dermal leishmaniasis | 0 | 0 | 0 |
| Onchocerciasis | 0 | 0 | 0 |

with a combination of rifampicin and clarithromycin according to WHO recommendations [26]. Category 2 and 3 cases were treated at the CDTLUB of Pobè, due to the size of the lesions and the need for a physiotherapist to prevent disability, a surgeon for lesion debridement and skin grafts, and nutritional and psychosocial care. Category 1 cases were managed at decentralized treatment sites run by the CDTLUB in the community, in collaboration with nurses from district health centers. The management of patients with Buruli ulcer in the community was managed by a multidisciplinary team until complete healing was achieved.

Six cases of leprosy (15.0% of the skin NTDs) were diagnosed during the first period of mobile consultations (**Fig 4B**). In 2019, there were five cases of leprosy (3.6% of the c skin NTDs). Finally, in 2020 11 cases were detected (2.3% of the skin NTDs). Characteristics of patients presenting leprosy are presented in **Table 3**. Leprosy cases were treated according to WHO recommendations [27]. Multidrug treatment regimens consisting of combinations of rifampicin, clofazimine and dapsone were used, for six months in paucibacillary cases, and for 12 months in multibacillary cases. Treatment was initiated at the CDTLUB of Pobè and continued on an outpatient basis. Patients were made aware of the importance of treatment for preventing leprosy-related disabilities and the need to complete treatment. They also received psychosocial support.

In addition, 15 cases of lymphatic filariasis, were diagnosed, two during the first period of mobile consultations (5.0% of skin NTDs), 10 cases (7.2%) during the second period of mobile consultations and three (0.6%), during the third period of mobile consultations. All cases were diagnosed during a phase in which complications had already occurred and they were managed by the CDTLUB of Pobè. They were systematically treated with a combination of albendazole and ivermectin [28] and were advised on how to care for their limbs and prevent complications, such as chronic ulcers, leg erysipelas, and secondary bacterial infections.

One patient (0.7%) was diagnosed with mycetoma, during the second period of mobile consultations (**Fig 4C**). He was treated with cotrimoxazole for one year.

Finally, all 571 patients diagnosed with scabies during the three periods of mobile consultations (**Fig 4D**) received free local antiseptics for total body cleansing twice daily (in the morning and evening), antihistamines to reduce the associated pruritus, ivermectin (dose according to weight) and a solution of benzyl benzoate for topical application.

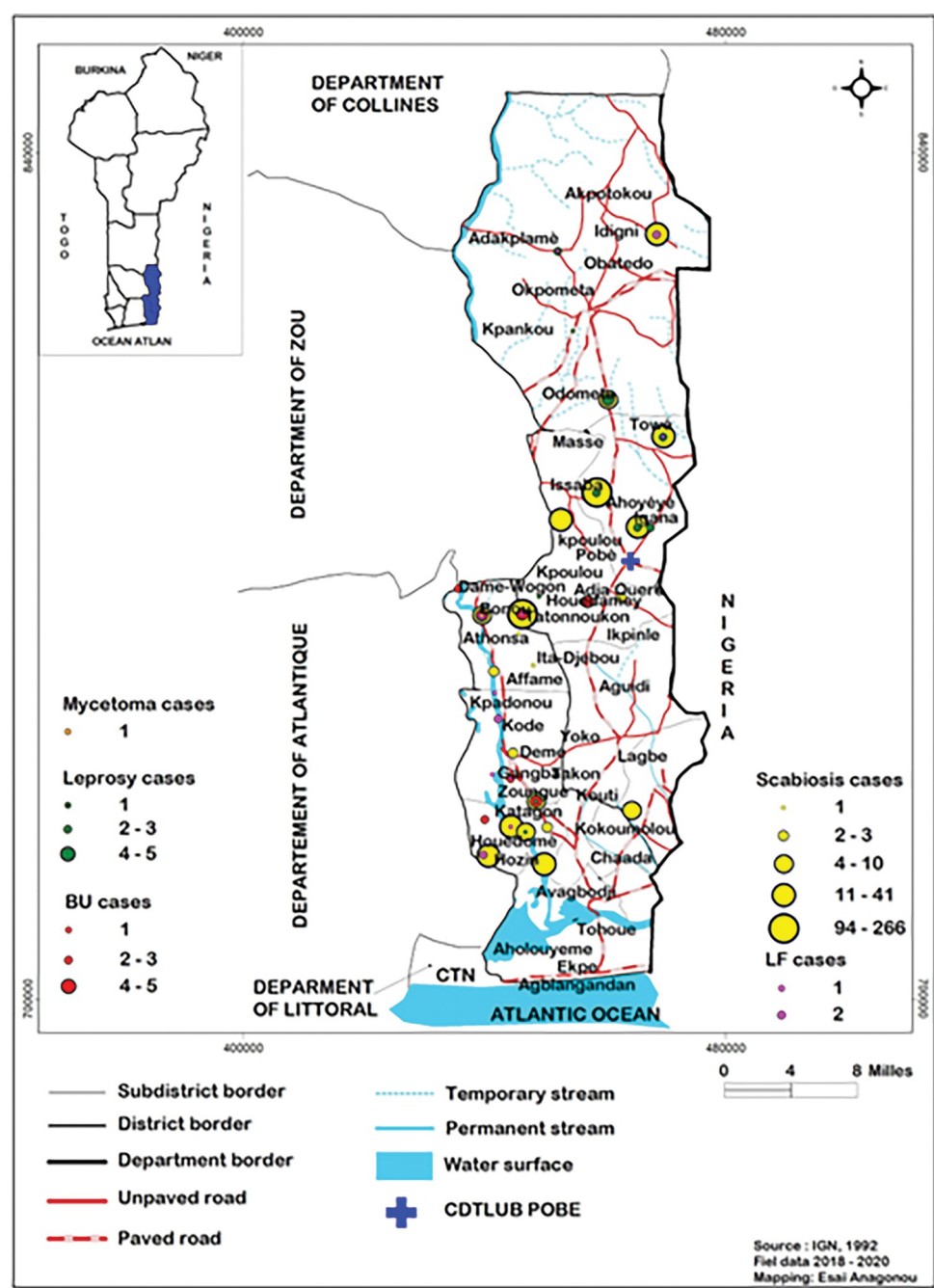

**Fig 3. Localization of sites of consultations with skin NTDs.** In green, leprosy cases, in red, Buruli ulcer (BU) cases, in yellow, scabies cases, in pink, lymphatic filariasis (LF) cases, in orange, mycetoma cases. Datas for the administrative boundary were provided by Ministry of Territory Planning of Benin (https://gadm.org/download_country_v3.html).

### 3.3 Diagnosis of other skin diseases

Most of the lesions diagnosed during the three periods of mobile consultations were of infectious origin: 67.9% in 2018, 72.4% in 2019 and 78.6% in 2020. Infections were mycotic in 46.6%, 52.9% and 52.6% of cases, respectively, in 2018, 2019 and 2020, respectively. A summary of all the skin diseases encountered is presented in **Table 4**. Ringworm and epidermophytia were the two major non-NTD skin diseases diagnosed, accounting for 18.6% and 12.7%

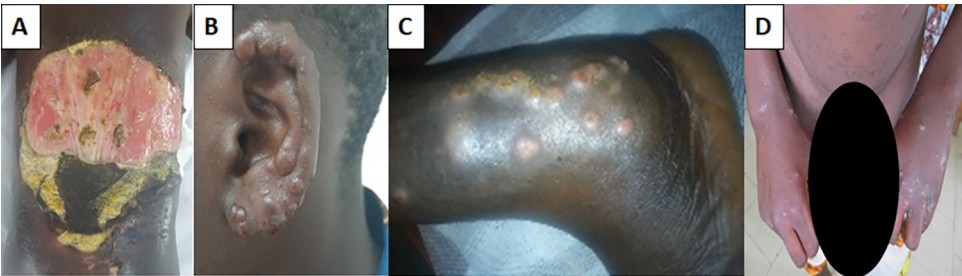

**Fig 4. Neglected tropical skin diseases seen during mobile consultations.** A) Buruli ulcer, B) leprosy, C) mycetoma, D) scabies. **Commentary about the illustrations and figures.** All the illustrations and figures are our own. They were taken during the implementation phase of this approach to integrate the fight against neglected tropical diseases with skin manifestations in the departments of Ouémé and Plateau in Benin from 2018 to 2020. We had taken them after obtaining informed consent and verbal authorization from all those who were considered. They were also informed and gave a verbal approval for the use of these iconographies in the framework of the valorization of this manuscript.

of all consultations in the first period, 18.4% and 23.7%, respectively, in the second period and 28.7% and 13.9%, respectively, in the third period. Other frequent non-NTD skin diseases included impetigo contagiosa, pityriasis versicolor, chronic ulcers and eczema. Most of the non-infectious skin diseases were immunoallergic conditions (**Table 4**).

Depending on the type of skin dermatosis identified, its severity and, above all, the availability of an adequate technical platform, patients received outpatient medical treatment or were referred to a specialist structure.

## IV Discussion

Skin NTDs are among the most prevalent infectious diseases in tropical and subtropical regions. They are difficult to manage due to the similarity of their clinical signs, their frequent co-endemicity and their occurrence in people living in poor regions with little or no access to healthcare. Integrated approaches are required to reduce the transmission of these diseases and to ensure that patients are managed earlier. In this context, we decided to develop a system of mobile consultations in two *départements* of Benin in which Buruli ulcer and leprosy are known to be highly endemic.

Our results indicate that these mobile consultations were highly successful. Indeed, in addition to the diagnosis and treatment of 646 cases of skin NTDs, the medical staff examined more than 5,200 people carrying various skin affections always with the aim of no exclusion or discrimination. The consultations were realised in respect of the gender approach, of the human person and of his dignity. Management of other skin diseases was also performed as optimally as possible and, if needed, the patients were referred to specialists. Furthermore, these consultations allowed us to see former patients treated at CDTLUB, pobè, allowing us to verify their scars and reinforcing our presence with these populations.

The mobile consultations provided useful epidemiological information about skin NTDs and other skin diseases in Plateau and Ouémé. This study provided us with a precise idea of the distributions of the skin diseases encountered, by administrative area and the sociodemographic and clinical characteristics of patients. Our study also provided a general overview of the most common skin NTDs and their stage of progression in the population. Similar experiments in the implementation of integrated management for skin NTDs have been performed in other countries, including Côte d'Ivoire, where Koffi et al. screened 2,310 individuals with skin lesions and diagnosed Buruli ulcer, leprosy and yaws in three health districts co-endemic for Buruli ulcer and leprosy [29].

**Table 3. Characteristics of patients diagnosed with Buruli ulcer and leprosy during the three periods of mobile consultations held in the Plateau and Ouémé in Benin.**

| | Buruli ulcer | Leprosy |
|---|---|---|
| **Total number of cases** | **37** | **22** |
| **≤ 15 years / > 15 years** no. (%) | 14 (37.8) / 23 (62.2) | 4 (18.2) / 18 (81.8) |
| **Sex ratio M/F** [no. (%)] | 0.762 [16/21 (43.2/ 56.8)] | 1.2 [12/10 (54.5/45.5)] |
| **Profession** no. (%) | | |
| Learner | 13 (35.1) | 2 (9.1) |
| Merchant | 5 (13.5) | 2 (9.1) |
| Farmer | 8 (21.6) | 13 (59.1) |
| No profession | 1 (2.7) | - |
| Artisan | 5 (13.5) | 2 (9.1) |
| Household | 5 (13.5) | 3 (13.6) |
| **Ethnic group** no. (%) | | |
| Wémé and related | 26 (70.3) | 3 (13.6) |
| Holly | 8 (21.6) | 14 (63.6) |
| Nagot and related | 1 (2.7) | 5 (22.8) |
| Fon and related | 2 (5.4) | - |
| **Educational level** no. (%) | | |
| Educated | 13 (35.1) | 2 (9.1) |
| Not educated | 24 (64.9) | 20 (90.9) |
| **Socio-economic level** no. (%) | | |
| Medium | 1 (2.7) | 6 (27.3) |
| Low | 36 (97.3) | 16 (72.7) |
| **Municipality** no. (%) | | |
| Pobè | 4 (10.8) | 9 (40.9) |
| Kétou | - | 9 (40.9) |
| Dangbo | 10 (27.0) | 2 (9.1) |
| Adjohoun | 7 (18.9) | 1 (4.6) |
| Adja-Ouèrè | 8 (21.6) | 1 (4.6) |
| Bonou | 7 (18.9) | - |
| Akpro-Missérété | 1 (2.7) | - |
| **Bascilloscopy** no. (%) | | |
| Positive | 11 (29.7) | 13 (59.1) |
| Negative | 26 (70.3) | 9 (40.9) |
| **Buruli ulcer lesion type** no. (%) | Ulceration 23 (62.2) Plaque 6 (16.2) Nodule 4 (10.8) Oedema 4 (10.8) | |
| **Buruli ulcer WHO classification** no. (%) | | |
| Category I | 17 (46.0) | |
| Category II | 11 (29.7) | |
| Category III | 9 (24.3) | |
| **Leprosy clinical classification** no. (%) | | |
| Grade 2 infirmity | | 12 (54.6) |
| Distribution | | ≤ 15 years 1 (4.5) / Women 5 (22.7) / Men 6 (27.3) |
| **Leprosy microbiological classification** no. (%) | | |
| Multibacillary | | 16 (72.7) |
| Paucibacillary | | 6 (27.3) |

Our study highlighted several advantages of this approach. First, we showed that mobile consultations facilitated the on-site treatment of patients living in remote villages. This strategy helped to reduce the burden of diseases that remained widespread. Second, this study led to the detection of a large number of skin diseases, including neglected diseases that had not been declared to health centers and were particularly widespread in remote areas. It therefore

**Table 4. Skin diseases observed during mobile consultations in Plateau and Ouémé in Benin.**

| Skin disease | Period 1 (Nov 6 to Dec 20, 2018) No. of cases (%) Total no. 607 | Period 2 (May 2 to Aug 17, 2019) No. of cases (%) Total no. 2,444 | Period 3 (Dec 1 to 21, 2020) No. of cases (%) Total no. 2,216 |
|---|---|---|---|
| **Infectious skin diseases** | **412 (67.9)** | **1,769 (72.4)** | **1,742 (78.6)** |
| *Mycotic dermatosis* | *283 (46.6)* | *1,293 (52.9)* | *1,166 (52.6)* |
| Ringworm scalp | 113 (18.6) | 449 (18.4) | 634 (28.6) |
| Epidermophytia | 77 (12.7) | 580 (23.7) | 308 (13.9) |
| Pityriasis versicolor | 55 (9.1) | 124 (5.1) | 122 (5.5) |
| Intertrigo | 16 (2.6) | 71 (2.9) | 42 (1.9) |
| Vulvovaginitis | 10 (1.7) | 21 (0.9) | 38 (1.7) |
| Onychomycosis | 8 (1.3) | 44 (1.8) | 20 (0.9) |
| Seborrheic dermatitis | 2 (0.3) | - | 2 (0.1) |
| Nappy rash | 2 (0.3) | 4 (0.1) | - |
| *Bacterial dermatoses* | *96 (15.8)* | *342 (14.0)* | *110 (5.0)* |
| Impetigo contagiosa | 65 (10.7) | 187 (7.7) | 74 (3.3) |
| Leprosy | 6 (1.0) | 5 (0.2) | 11 (0.5) |
| Buruli ulcer | 3 (0.5) | 28 (1.2) | 6 (0.3) |
| Yaws | - | 2 (0.1) | - |
| Mycetoma | - | 1 (0.0) | - |
| Furuncle | 11 (1.8) | 71 (2.9) | 8 (0.4) |
| Osteomyelitis | 11 (1.8) | 22 (0.9) | 9 (0.4) |
| Erysipelas | - | 26 (1.0) | 1 (0.1) |
| Gonococcal urethritis | - | - | 1 (0.1) |
| *Parasitic dermatoses* | *31 (5.1)* | *104 (4.3)* | *454 (20.5)* |
| Scabies | 29 (4.8) | 93 (3.8) | 450 (20.3) |
| Lymphatic filariasis | 2 (0.3) | 10 (0.4) | 3 (0.1) |
| Larva migrans | - | 1 (0.1) | 1 (0.1) |
| *Viral dermatoses* | *2 (0.3)* | *30 (1.2)* | *12 (0.6)* |
| Herpes simplex | 2 (0.3) | 11 (0.5) | 2 (0.1) |
| Herpes zoster | - | 13 (0.5) | 8 (0.3) |
| Wart | - | 4 (0.2) | 1 (0.1) |
| Condyloma | - | 2 (0.0) | 1 (0.1) |
| **Non-infectious skin diseases** | **195 (32.1)** | **675 (27.6)** | **474 (21.4)** |
| *Immunoallergic dermatoses* | *89 (14.7)* | *287 (11.7)* | *287 (13.0)* |
| Prurigo | 39 (6.4) | 53 (2.2) | 119 (5.4) |
| Eczema | 23 (3.8) | 108 (4.4) | 88 (4.0) |
| Atopic dermatitis | 15 (2.5) | 87 (3.6) | 43 (1.9) |
| Palmoplantar keratoderma | 7 (1.2) | 23 (0.9) | 8 (0.4) |
| Urticaria | 4 (0.6) | 13 (0.5) | 24 (1.1) |
| Keratosis pilaris | 1 (0.2) | 3 (0.1) | 1 (0.1) |
| Fixed pigmented erythema | - | - | 4 (0.2) |
| *Tumoral dermatoses* | *38 (6.3)* | *93 (3.8)* | *44 (2.0)* |
| Fibrosing folliculitis | 19 (3.1) | 19 (0.8) | 20 (0.9) |
| Keloid | 5 (0.8) | 46 (1.9) | 11 (0.5) |
| Lipoma | 5 (0.8) | 16 (0.6) | 6 (0.3) |
| Synovial cyst | 1 (0.2) | 7 (0.3) | 5 (0.2) |
| Hamartoma | 3 (0.5) | 1 (0.0) | 2 (0.1) |
| Metastatic tumor | 4 (0.7) | 4 (0.2) | - |

*(Continued)*

**Table 4.** (Continued)

| Skin disease | Period 1 (Nov 6 to Dec 20, 2018) No. of cases (%) Total no. 607 | Period 2 (May 2 to Aug 17, 2019) No. of cases (%) Total no. 2,444 | Period 3 (Dec 1 to 21, 2020) No. of cases (%) Total no. 2,216 |
|---|---|---|---|
| Syringoma | 1 (0.2) | - | - |
| *Inflammatory dermatoses* | *32 (5.3)* | *60 (2.5)* | *57 (2.6)* |
| Acne | 14 (2.3) | 12 (0.5) | 18 (0.8) |
| Sudamina (miliaria) | 11 (1.8) | 36 (1.5) | 17 (0.8) |
| Lichen | 4 (0.7) | 8 (0.3) | 15 (0.7) |
| Alopecia areata | 2 (0.3) | 0 (0) | 3 (0.1) |
| Psoriasis | 1 (0.2) | 4 (0.2) | 4 (0.2) |
| *Ulcerative dermatoses* | *27 (4.4)* | *212 (8.7)* | *71 (3.2)* |
| Chronic non-BU ulcer | 24 (3.9) | 212 (8.7) | 65 (2.9) |
| Plantar perforating pain | 3 (0.5) | - | 6 (0.3) |
| *Genodermatoses* | *6 (1.0)* | *9 (0.4)* | *4 (0.2)* |
| Neurofibromatosis | 3 (0.5) | 8 (0.3) | 3 (0.1) |
| Congenital ichthyosis | 2 (0.3) | 1 (0.1) | 1 (0.1) |
| Ainhum disease | 1 (0.2) | - | - |
| *Autoimmune dermatoses* | *2 (0.3)* | *11 (0.5)* | *3 (0.1)* |
| Vitiligo | 2 (0.3) | 10 (0.4) | 2 (0.1) |
| Lupus erythematosus | - | 1 (0.1) | 1 (0) |
| *Vascular dermatoses* | *1 (0.2)* | *3 (0.1)* | *2 (0.1)* |
| Tuberous hemangioma | 1 (0.2) | 3 (0.1) | 2 (0.1) |
| *Other dermatoses* | - | - | *6 (0.3)* |
| Pellagra | - | - | 2 (0.1) |
| Scrotal calcinosis | - | - | 2 (0.1) |
| Exogenous ochronosis | - | - | 2 (0.1) |

provided new data that could be used to raise awareness among associations and government institutions, because the data obtained in hospitals and health centers only partially reflects the reality on the ground and does not accurately represent the true burden of these diseases. Third, we observed an increase in the number of people agreeing to take part in mobile consultations between periods 1 and 3. This may be due to an increase in awareness about these diseases and the availability of free treatment for patients suffering from them. A relationship of trust was also established between the population and the medical team over the three periods of consultation.

Mobile consultations have brought a change of paradigm in the care of patients. Indeed, usually, it was the patient who went to the doctor but through this approach, it was the caregivers who went to the patients. Thus, health care was made with quality, was accessible to these populations in remote areas and participated in reducing the burden of morbidity linked to NTDs in these localities. This approach also was developed in a more human approach since taking into account the low standard of living of the population, consultation were made free and the management of cases was ensured. Finally, human, material, financial and time resources were mutualized and rationalized to screen for several skin NTDs and many other skin diseases in order to be as most efficient as possible.

Buruli ulcer diagnosis by PCR was introduced in Benin in 2004 following an increase in the number of cases in the country [30,31]. This infectious disease is characterized by painless skin lesions that can lead to scarring, contracture deformities, amputations and disabilities if left

untreated [31]. Along with Côte d'Ivoire, Ghana and Cameroon, Benin is one of the four African countries in which Buruli ulcer is most endemic. Between 2006 and 2015, the CDTLUB of Pobè diagnosed and treated 1,365 patients, 69% of whom were from Ouémé, with the other 31% from Plateau [32]. During the mobile consultations in this study, 37 patients were diagnosed and treated. Most of the cases of Buruli ulcer detected during the three consultation periods were from Ouémé, which is crossed by the River Ouémé. There is an alternation of dry and wet seasons in this *département*. During wet seasons, a large area around the river is flooded, resulting in large expanses of stagnant water, which constitute the main ecological niche of the bacillus, creating an environment in which the disease is predominant [33].

Scabies remained the leading skin NTD, accounting for the highest proportion of cases identified, over these three years of mobile consultations. This disease constitutes a real public health problem because of its high prevalence, and its high morbidity in children [34]. For the 5,267 people presenting skin diseases during the three years of mobile consultations, the overall prevalence of scabies was 10.8%, with most cases occurring in children under the age of 15 years. In 2020, the consultation team came across community cases of scabies, initially in the Pobè municipal area and, more specifically, in the district of Issaba, and subsequently in the Adja-Ouèrè municipal area, in which the incidence of scabies was very high. Almost everyone, including the healthcare agents, had scabies in these areas. This observation led us to alert the local health authorities and the Ministry of Health, which organized mass treatment with ivermectin and made benzyl benzoate solution available to healthcare facilities to ensure correct case management.

Associations of scabies, impetigo and superficial mycoses are often attributed to developing countries, a lack of hygiene being the factor common to these three infections in the vulnerable populations of these countries [35,36]. We also found the prevalence of these three infections to be high. A specific strategy should therefore be implemented to target dermatoses of this type by raising the awareness of the population concerning the importance of maintaining good physical hygiene.

We also detected 22 cases of leprosy during the three years of mobile consultations, this disease being the third most frequent after scabies and Buruli ulcer. Leprosy is a chronic disease caused by a slow-growing bacterium, *Mycobacterium leprae*. The consultations in 2018 and 2020 detected the largest numbers of leprosy cases (6 cases in 2018 and 11 cases in 2020), corresponding to a prevalence of 0.08 and 0.11%, respectively. These values indicate that Benin remains among the countries considered to have eliminated leprosy, as the prevalence of this disease remains below 1 per 10,000 inhabitants. Nevertheless, this disease remains a major health challenge due to the large proportion of multibacillary cases and the increase in the number of individuals newly registered with grade 2 disabilities. In this context, it is important to strengthen the epidemiological surveillance of leprosy and to encourage its early detection in populations at risk, by adopting new strict strategies, such as the implementation of diagnostic tools for detection of the bacillus and for the detection of resistance genes.

The declaration of skin NTDs to health centers is not compulsory, accounting for the often poor knowledge of healthcare workers concerning these diseases [37]. In addition to the large distances between health centers and villages, the poor quality of roads, the lack of transport vehicles, and the high rates of very severe poverty in these populations [38], the painless nature of certain types of skin NTDs, such as Buruli ulcer, mycetoma, cutaneous leishmaniasis, yaws and some forms of leprosy, is an important factor underlying their neglected nature [39–43]. If untreated, these diseases progress to severe forms, leading to permanent physical disability altering quality of life (work, education, mental state, etc.), highlighting the importance of mobile consultations for managing patients.

The consultation team encountered several problems that lowered its efficiency. For example, it was difficult to pick up the patients affected by diseases in very remote regions, particularly as the consultations took place during the rainy season, when it was difficult to cross rivers without motorized boats. Moreover, shortages of medical consumables for basic necessities, such as antifungal drugs (oral and topical), antiparasitic agents and topical corticosteroids for patient care were also identified as problematic and a source of concern for the team. Finally, some local actors were found to be less implicated in the project and a lack of means for raising awareness was also noted.

Based on the results obtained, those involved in the epidemiological surveillance of skin NTDs are faced with a number of challenges. Action will be required to meet these challenges without jeopardizing the chances of achieving the objectives of the new roadmap developed by the WHO for 2030 [44]. It will be important:

i. To ensure the maintenance of the achievements attained by implementing this approach in Ouémé and Plateau through social mobilization, training of health workers on screening and integrated management of NTDs, wound care and encouraging the routinization of this approach to take into account a large number of people,

ii. To lobby political and administrative authorities at various levels to get them to increase the budget allocated to the fight against tropical skin diseases. Efforts should be directed towards encouraging the construction of sociosanitary infrastructures in these isolated rural areas to facilitate the access of these populations to primary healthcare and to meeting the objectives of sustainable development (ensuring universal access to medical coverage and health services). Conversely, it will also be important to integrate activities for combating skin NTDs into the primary healthcare activities offered to populations by peripheral healthcare centers. For example, when performing vaccinations or routine prenatal consultations, healthcare workers should systematically look for a change in the color or relief of the skin,

iii. To intensify activities for raising awareness, to change the behavior of the population in terms of risky behaviors contributing to the occurrence of these diseases,

iv. To implement activities designed to promote health,

v. To expand the scope of implementation of mobile clinics to other regions of Benin in which these diseases are endemic. If resources permit, non-endemic areas that have remained "silent" epidemiologically should also be considered.

Another crucial transverse component must also be added to all these activities to fight disease or reduce the burden if skin NTDs: water, hygiene and sanitation, as most of these diseases are favored by a lack of hygiene and environments conducive to their occurrence. This is why one of the strategic challenges in the fight against NTDs in Ouémé and Plateau will be making clean drinking water available in remote areas. We also need to improve the access of the population to healthcare facilities, to construct latrines, and to destroy the breeding grounds of disease. In other words, we need to clean up the environment to make it healthy, viable and livable, to guarantee the health of the population. The intervention and involvement of other health-related sectors will be essential to achieve this goal. The challenge will therefore be to federate the actions of the Ministry of Health, the Ministry of the Environment and the Living Environment, the Ministry of Water and Electricity, the Ministry of Education (to improve the level of education of the population and to promote the enrolment of all school-age children), the Ministry of Agriculture and Fisheries and the Ministry of Finance (which could promote small income-generating activities to lift the population out of poverty). All of

these sectors must act holistically on the other determinants of health. In other words, the challenge will be to combine the integrated fight against these diseases with the implementation of the "One Health" concept.

## V. Conclusion

Mobile consultations are a highly efficient strategy for the integrated screening of skin diseases, particularly NTDs with skin manifestations, which are often diagnosed late. Sustaining these activities could help to reduce the burden of NTDs and contribute to the achievement of the objectives of the new WHO roadmap developed for the 2030 horizon.

## Acknowledgments

We thank the Coordinator of the National Program for the Control of Leprosy and Buruli Ulcer in Benin, all academic institutions affiliated with this research (University of Angers, University of Abomey Calavi; the Inter-Faculty Training Center for Research in the Environment for Sustainable Development; the Regional Institute of Public Health Comlan Alfred Quenum of Ouidah) and all technical partners, including Fondation Raoul Follereau (http://www.raoul-follereau.org), the World Health Organization and the ANESVAD Foundation for their contributions towards efforts to combat neglected tropical diseases more effectively in Benin. INSERM supports the collaboration through the label IRP (International Research Program) "Prevent-BU".

## Author Contributions

**Conceptualization:** Jean Gabin Houezo, Béatriz Gomez, Anna Gine, Ghislain Emmanuel Sopoh, Roch Christian Johnson.

**Data curation:** Ronald Sètondji Gnimavo, Espoir Sodjinou, Akimath Habib, Line Ganlonon, Eric Claco, Irvine Agoundoté.

**Formal analysis:** Ronald Sètondji Gnimavo, Faraj Fajloun, Charbel Al-Bayssari.

**Methodology:** Ronald Sètondji Gnimavo, Espoir Sodjinou, Akimath Habib, Line Ganlonon, Eric Claco, Irvine Agoundoté, Alexandra Boccarossa, Elie Hajj Moussa.

**Supervision:** Ronald Sètondji Gnimavo, Faraj Fajloun, Odile Adjouavi Houngbo, Esaï Gimatal Anagonou, Chabi Alphonse Olaniran Biaou, Adjimon Gilbert Ayélo, Jean Gabin Houezo, Béatriz Gomez, Anna Gine, Roch Christian Johnson, Marie Kempf.

**Writing – original draft:** Ronald Sètondji Gnimavo, Faraj Fajloun, Charbel Al-Bayssari, Estelle Marion, Roch Christian Johnson, Marie Kempf.

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
