## [Decision Letter · Decision Letter 0]

6 Jan 2023

Dear Pr. Kempf,

Thank you very much for submitting your manuscript "Importance of mobile consultations in the screening and treatment of neglected tropical skin diseases in Benin." for consideration at PLOS Neglected Tropical Diseases. As with all papers reviewed by the journal, your manuscript was reviewed by members of the editorial board and by several independent reviewers. In light of the reviews (below this email), we would like to invite the resubmission of a significantly-revised version that takes into account the reviewers' comments. 

We cannot make any decision about publication until we have seen the revised manuscript and your response to the reviewers' comments. Your revised manuscript is also likely to be sent to reviewers for further evaluation.

Sincerely,

Ahmed Hassan Fahal, FRCS, FRCSI, FRCSG, MS, MD, FRCP(London)

Academic Editor

Dileepa Ediriweera

Section Editor

Reviewer's Responses to Questions

**Key Review Criteria Required for Acceptance?**

**Methods**

-Are the objectives of the study clearly articulated with a clear testable hypothesis stated?

-Is the study design appropriate to address the stated objectives?

-Is the population clearly described and appropriate for the hypothesis being tested?

-Is the sample size sufficient to ensure adequate power to address the hypothesis being tested?

-Were correct statistical analysis used to support conclusions?

-Are there concerns about ethical or regulatory requirements being met?

Reviewer #1: Is mobile consultation the best descriptor to introduce the concept as many will think that this is referring to the use of mobile phone technology to deliver these objectives – see examples in Mali and Botswana for instance. ? Perhaps “consultations using mobile teams” would be better in the title and when first mentioned in the text. If you state this in the title then you can use “mobile teams” in the rest of the manuscript

 The assertion that WHO was encouraging integration for control strategies for NTDs with skin manifestations from 1997 is not accurate – this is a much more recent strategy. 

Line 128. Please provide figures that show the high levels of endemicity referred to. 

It would be useful, for the authors to state clearly the differences between this study and the earlier Benin study – their reference 4 . How does it differ – some of the authors are the same ?

What training/preparation was given to the multidisciplinary team. Some were specialists but most were not – so it would be important to show how the teams worked. Did they use written material, diagnostic Apps or field training ?

Reviewer #2: The objectives of the study is not clearly stated and it would be good for the author to clearly state the objectives. Hypthesis they wish to test has also to be clear.

The study design seem ok but should be linked to clearly stated objectives. 

Sample size is not appropriate for the observational study done so can be skipped.

Ethical approvals were obtained for the study

**Results**

-Does the analysis presented match the analysis plan?

-Are the results clearly and completely presented?

-Are the figures (Tables, Images) of sufficient quality for clarity?

Reviewer #1: Can the authors comment on the missing cases ie the patients who were not interested in coming to the clinics – was this a large number ?

Was any feedback provided to the participating communities after the clinics ? Do you have any more information on the outcomes of the much higher number of cases who had skin diseases which were not NTDs. ie Table 4. The satisfaction of these patients is important to the long term sustainability of this work as it may determine whether people will come forward in future for these mobile skin clinics

Table 4

What is epidermophytia ? – how does it differ from ringworm . 

Presumably the mycetoma was bacterial or was it fungal in which case it should be with the fungal infections 

What is fibrosing folliculitis – is this scarring alopecia in which case should it be tumoral, or is it acne keloid ? 

Hot abscess – abscess or furunculosis ?

Herpes – presumably Herpes simplex

Sudamina – miliaria

Reviewer #2: The analysis is fine and results clearly presented.

Images can be improved - quality images.

**Conclusions**

-Are the conclusions supported by the data presented?

-Are the limitations of analysis clearly described?

-Do the authors discuss how these data can be helpful to advance our understanding of the topic under study?

-Is public health relevance addressed?

Reviewer #1: From line 449 onwards there are a series of actions listed that would help to sustain this work – how many of these have been implemented ? 

What impact has COVID 19 had on this project ?

Reviewer #2: Discussions be limited to the data/results. Extrapolations not supported by the data are not needed. Extensive discussions of intersectoral collaboration are not needed. It is easy to say but difficult to do.

The strategy of mobile outreach has been used for decades but to see it used in Benin in the context of skin NTDs surveillance and diagnosis has provided good data on prevalent skin diseases in the surveyed areas. Large number of skin diseases are not skin NTDs and their management challenges should be discussed in detail.

**Editorial and Data Presentation Modifications?**

Reviewer #1: Please check spelling and references as there are some errors

Reviewer #2: (No Response)

**Summary and General Comments**

Reviewer #1: Overall this is interesting work. It would be good to hear more about how the authors intend to ensure sustainability of this work.

Reviewer #2: Please use Skin NTDs rather than cutaneous NTDs

Change cutaneous to skin

Get a native English speaker to review the paper

Check reference and use the most recent references to NTDs.

Line 111: Please check the statement on eradication of the three diseases carefully. Please refer to the NTD Road Map 2021-2030. Buruli ulcer is for control, leprosy for elimination and yaws for eradication.

Line 179: individuals with dermatological lesions not regions

Line 195: Yaws and not yaw

Line 194: Please provide reference or revise the sentence.

Line 197: BU swabs are taken from the undermined edges of the ulcer (please make this clear)

Line 277: Use male or female instead of man and woman

Line 281: Use debridement instead of lesion trimming

Line 285: What multidisciplinary team was in the community?

Line 361: How did the strategy helped to reduce the burden of these diseases? 

Line 480-493: Discussions outside the data generated. Kindly discuss the results and avoid extrapolation to what is not provided by data collected. Inter-sectorial collaboration is good but very hard to achieve.

PLOS authors have the option to publish the peer review history of their article (what does this mean?). If published, this will include your full peer review and any attached files.

Reviewer #1: No

Reviewer #2: No
---

## [Decision Letter · Decision Letter 1]

14 Mar 2023

Dear Pr. Kempf,

Thank you very much for submitting your manuscript "Importance of consultations using mobile teams in the screening and treatment of neglected tropical skin diseases in Benin" for consideration at PLOS Neglected Tropical Diseases. As with all papers reviewed by the journal, your manuscript was reviewed by members of the editorial board and by several independent reviewers. The reviewers appreciated the attention to an important topic. Based on the reviews, we are likely to accept this manuscript for publication, providing that you modify the manuscript according to the review recommendations. 

Sincerely,

Ahmed Hassan Fahal, FRCS, FRCSI, FRCSG, MS, MD, FRCP(London)

Academic Editor

Dileepa Ediriweera

Section Editor

Reviewer's Responses to Questions

**Key Review Criteria Required for Acceptance?**

**Methods**

-Are the objectives of the study clearly articulated with a clear testable hypothesis stated?

-Is the study design appropriate to address the stated objectives?

-Is the population clearly described and appropriate for the hypothesis being tested?

-Is the sample size sufficient to ensure adequate power to address the hypothesis being tested?

-Were correct statistical analysis used to support conclusions?

-Are there concerns about ethical or regulatory requirements being met?

Reviewer #1: Fine

**Results**

-Does the analysis presented match the analysis plan?

-Are the results clearly and completely presented?

-Are the figures (Tables, Images) of sufficient quality for clarity?

Reviewer #1: The authors have addressed most of my comments. I still have concerns about the words used in the table to describe skin disease as they wont be understood by many dermatologists

Ringworm is used everywhere else to describe all dermatophyte infections - scalp , body etc. So to make this more widely understandable I would use the words Ringworm (scalp) and Ringworm ( other sites )

Thank you for clarifying the meaning of hot access. The English dermatological term used for this is a Boil or Furuncle - this again would make the diagnosis easier to understand

**Conclusions**

-Are the conclusions supported by the data presented?

-Are the limitations of analysis clearly described?

-Do the authors discuss how these data can be helpful to advance our understanding of the topic under study?

-Is public health relevance addressed?

Reviewer #1: Fine

**Editorial and Data Presentation Modifications?**

Reviewer #1: Fine

**Summary and General Comments**

Reviewer #1: Fine

PLOS authors have the option to publish the peer review history of their article (what does this mean?). If published, this will include your full peer review and any attached files.

Reviewer #1: No

Figure Files:

Data Requirements:

Reproducibility:

References

---

## [Editor Report · Decision Letter 2]

15 Apr 2023

Dear Pr. Kempf,

We are pleased to inform you that your manuscript 'Importance of consultations using mobile teams in the screening and treatment of neglected tropical skin diseases in Benin' has been provisionally accepted for publication in PLOS Neglected Tropical Diseases.

Best regards,

Ahmed Hassan Fahal, FRCS, FRCSI, FRCSG, MS, MD, FRCP(London)

Academic Editor

Dileepa Ediriweera

Section Editor

---

## [Editor Report · Acceptance letter]

4 May 2023

Dear Pr. Kempf,

We are delighted to inform you that your manuscript, "Importance of consultations using mobile teams in the screening and treatment of neglected tropical skin diseases in Benin," has been formally accepted for publication in PLOS Neglected Tropical Diseases.

Best regards,

Shaden Kamhawi

co-Editor-in-Chief

Paul Brindley

co-Editor-in-Chief
